# MiR-486-5p Targets CD133+ Lung Cancer Stem Cells through the p85/AKT Pathway

**DOI:** 10.3390/ph15030297

**Published:** 2022-02-28

**Authors:** Massimo Moro, Orazio Fortunato, Giulia Bertolini, Mavis Mensah, Cristina Borzi, Giovanni Centonze, Francesca Andriani, Daniela Di Paolo, Patrizia Perri, Mirco Ponzoni, Ugo Pastorino, Gabriella Sozzi, Mattia Boeri

**Affiliations:** 1Tumor Genomics Unit, Department of Research, Fondazione IRCCS Istituto Nazionale dei Tumori, Via Venezian 1, 20133 Milan, Italy; massimo.moro@istitutotumori.mi.it (M.M.); orazio.fortunato@istitutotumori.mi.it (O.F.); giulia.bertolini@istitutotumori.mi.it (G.B.); mavis.mensah@uhcw.nhs.uk (M.M.); cristina.borzi@istitutotumori.mi.it (C.B.); giovanni.centonze@istitutotumori.mi.it (G.C.); francesca.andriani@ens-lyon.fr (F.A.); 2Virology and Molecular Pathology Department, University Hospital Coventry and Warwickshire, Coventry CV2 2DX, UK; 3First Pathology Division, Department of Pathology and Laboratory Medicine, Fondazione IRCCS Istituto Nazionale dei Tumori di Milano, 20133 Milan, Italy; 4Institute de Genomique Fonctionnelle de Lyon, CNRS UMR 5242, Ecole Normale Superieure de Lyon, Universite Claude Bernard Lyon 1, F-69364 Lyon, France; 5Laboratory of Experimental Therapies in Oncology, IRCCS Istituto Giannina Gaslini, 16147 Genoa, Italy; danieladipaolo@gaslini.org (D.D.P.); patriziaperri@gaslini.org (P.P.); mircoponzoni@gaslini.org (M.P.); 6Nuclear Medicine Unit, Santa Corona Hospital, 17027 Pietra Ligure, Italy; 7Thoracic Surgery Unit, Fondazione IRCCS Istituto Nazionale dei Tumori, Via Venezian 1, 20133 Milan, Italy; ugo.pastorino@istitutotumori.mi.it

**Keywords:** microRNA, lung cancer, cancer stem cell

## Abstract

Despite improvements in therapies and screening strategies, lung cancer prognosis still remains dismal, especially for metastatic tumors. Cancer stem cells (CSCs) are endowed with properties such as chemoresistance, dissemination, and stem-like features, that make them one of the main causes of the poor survival rate of lung cancer patients. MicroRNAs (miRNAs), small molecules regulating gene expression, have a role in lung cancer development and progression. In particular, miR-486-5p is an onco-suppressor miRNA found to be down-modulated in the tumor tissue of lung cancer patients. In this study, we investigate the role of this miRNA in CD133+ lung CSCs and evaluate the therapeutic efficacy of coated cationic lipid-nanoparticles entrapping the miR-486-5p miRNA mimic (CCL-486) using lung cancer patient-derived xenograft (PDX) models. In vitro, miR-486-5p overexpression impaired the PI3K/Akt pathway and decreased lung cancer cell viability. Moreover, miR-486-5p overexpression induced apoptosis also in CD133+ CSCs, thus affecting the in vivo tumor-initiating properties of these cells. Finally, we demonstrated that in vivo CCL-486 treatment decreased CD133+ percentage and inhibited tumor growth in PDX models. In conclusion, we provided insights on the efficacy of a novel miRNA-based compound to hit CD133+ lung CSCs, setting the basis for new combined therapeutic strategies.

## 1. Introduction

Lung cancer is still the main cause of cancer deaths worldwide, with 1.8 million deaths (18%) overall [1]. Despite improvements in early diagnosis with the advance of low-dose computed tomography (LDCT) screening trials and new therapeutic strategies, the 5-year survival rate in non-small-cell lung cancer (NSCLC) patients is estimated at around 21% in the U.S. and it drops dramatically to 6% for metastatic tumors [2].

The gold standard for early-stages tumors is surgical resection, alone or in combination with platinum-based doublet chemotherapy and radiation. Nonetheless, the CD133+ subset of tumor cells showed intrinsic chemoresistant properties such as high expression of drug efflux transporters, a relative quiescent status, and self-renewal ability [3,4]. In fact, according to the cancer stem cells (CSCs) theory, a small subpopulation of tumor-initiating cells with stem-like properties is responsible for the development and maintenance of the neoplasm [5]. Therefore, a deeper investigation of the molecular processes underlying lung cancer development and progression, with a particular focus on CSC biology, is critical to improve lung cancer management.

MicroRNAs are small, non-coding RNAs able to induce mRNAs degradation through the binding of complementary sequences of their targets. An altered expression of miRNAs has been described in several human malignancies, including lung cancer [6]. These small molecules are also involved in cell-to-cell communication since they can be released intact by cells encapsulated in microvescicles and perform their function in the recipient cells [7]. The revealing role of miRNAs as potential oncogenes and tumor suppressors has generated great interest and their characteristics make them potential therapeutic compounds [8].

Previously, we generated miRNA signatures with diagnostic and prognostic value in tumor, normal lung tissue, and plasma samples [9]. Indeed, we largely validated a panel of circulating miRNAs able to discriminate among lung cancer patients with the worst outcomes [10]. The study of several miRNAs composing the signature uncovered their involvement in lung cancer development and aggressiveness. Indeed, we demonstrated that miR-660 plays a crucial role in lung cancer cell survival by targeting the MDM2 mRNA and thus inhibiting tumor growth in xenograft mice models carrying p53 wild-type NSCLC [11]. More recently, it has been reported that high levels of miR-17 by targeting LKB1 identify NSCLC patients who are eligible for metabolic therapeutic strategy because they show sensitivity to energetic stress condition [12]. Moreover, the simultaneous modulation of miR-126-3p and miR-221-3p inhibits the phosphatase and tensin homolog (PTEN) and the phosphoinositide-3-kinase receptor 2 (PI3KR2), thus reducing metastatic dissemination of lung cancer cells [13].

Another interesting onco-suppressor miRNA down-modulated in lung cancer tissue and in plasma samples of patients with the worst outcome was miR-486-5p [9,14]. This miRNA may affect lung cancer cell proliferation, invasion, and metastasis and induce apoptosis in a TP53-dependent manner [15]. Among the main targets, mRNAs such as the forkhead box O1A (FOXO1A) [16], the Rho GTPase activating protein 5 (ARHGAP5) [17], members of the insulin-like growth factor (IGF) signaling, and the PI3K regulatory subunit 1-alpha (p85) were already described [15,18]. Interestingly, the CD133 glycoprotein can bind p85 at the level of the cellular membrane of CSCs in glioma models and subsequently activates the PI3K/Akt survivability signaling axis promoting the self-renewal and tumor formation of these cells [19]. On the other hand, the role of miR-486-5p on the control of CD133+ CSC has never been evaluated.

In this study, we wanted to elucidate the role of miR-486-5p on lung CD133+ CSCs by in vitro and in vivo experiments. By using coated cationic lipid-nanoparticles (CCL) entrapping the miR-486-5p miRNA mimic (CCL-486), we also evaluated the therapeutic efficacy of the compound using lung cancer patient-derived xenograft models (PDXs), which closely mirrors the main features of the parental human tumor [20].

## 2. Results

### 2.1. miR-486-5p Expression in Tumor Tissue and Plasma Samples of Lung Cancer Patients

Paired normal lung and tumor tissue samples collected from 45 NSCLC patients identified during two independent lung cancer screening trials were analyzed to evaluate miR-486-5p expression by RT-qPCR. No main differences were reported comparing the clinico-pathological characteristics of NSCLC patients in the two series, as reported in Table 1.

As for enrolment criteria of the screening trials, all patients were heavy smokers with Pack-Year >20 and older than 50. The majority of patients were males and diagnosed with stage I adenocarcinoma. Indeed, in both the series, lower levels of miR-486-5p were found in the tumor compared to paired normal lung tissue, irrespective of tumor stage (Figure 1).

### 2.2. miR-486-5p Overexpression Impairs the PI3K/AKT Pathway and Cell Viability

To evaluate the effects of miR-486-5p overexpression in lung cancer cell lines, the miRNA mimic (mim-486) as well as the control vector (ctrl-v) were transiently transfected in four commercial lung cancer cellular models. The effective miR-486-5p up-modulation was confirmed by RT-qPCR every 24 h up to 96 h (Appendix A). As already reported [15], by in vitro assays, the transfection of mim-486 in three TP53 WT cell lines (H460, LT73, and A549), but not in TP53 null cell line (H1299), induced cell apoptosis and reduced cell migration and invasion (Appendix A). On the other hand, in vivo xenograft models obtained by subcutaneous injection of transiently transfected cell lines in SCID mice revealed that mim-486 transfection induced a marked decreased in the tumor growth rate compared to controls in all xenograft models, including H1299 (Appendix A).

A significant difference in tumor growth was also observed in mim-486- versus ctrl-v-transfected H1299, suggesting a possible TP53-independent mechanism of action. Using the miRWalk database to uncover mRNA targets that could justify the discrepancies observed by in vitro and in vivo results, we focused our attention on p85, involved in the PI3K/Akt survivability pathway (Appendix A).

The effects of miR-486-5p overexpression on cell survivability were evaluated in lung cancer cell lines transfected with ctrl-v, mim-486, and the small interference RNA targeting p85 (siRNA-p85). To promote Akt phosphorylation and activate the survivability pathway, cells were then cultured with FBS free medium for 48 h to mimic starvation. By flow-cytometry, an overall increase in the percentage of 7AAD-/Annexin+ early apoptotic cells was observed when culturing them in starvation rather than in normal condition (Figure 2A,B). Moreover, a further significant increase in 7AAD-/Annexin+ cells in all the lung cancer cell lines was observed when transfecting the mim-486 or the siRNA-p85, including in the H1299 (Figure 2A,B). Western-blot analyses were further performed in protein extracted from starved cell lines to evaluate the PI3K/Akt pathway status 48 h after mim-486 transfection. The results reported a reduction in p85 protein levels, as well as the downstream phosphorylation of Akt in all mim-486- or siRNA-p85-transfected cell lines (Figure 2C).

### 2.3. miR-486-5p Overexpression Affects CD133+ Lung Cancer Stem Cell Survivability

Previous data in glioma CSC demonstrated that CD133 protein interact with p85 at the level of the cellular membrane to induce Akt phosphorylation [19]. The expression profiles of miR-486-5p and p85 in the four parental lung cancer cell lines, as well as the percentage of CD133+ cells evaluated by flow-cytometry, suggest the presence of the cross-talk between the miR-486-5p/p85 axis and the CSC phenotype also in these models (Appendix A). Moreover, immuno-fluorescence experiments, carried out on the primary adenocarcinoma tissue and in the corresponding cell line established in our laboratory (LT73), indicate a possible co-localization of p85 and CD133 at the level of the cellular membrane in lung CD133+ cells (Figure 3A,B). Conversely, in CD133neg cells, p85 localizes exclusively in the cytoplasm.

To better assess if the silencing of p85 could affect the CD133+ CSC compartment, the four lung cancer cell lines were transiently transfected with mim-486 or siRNA-p85 by flow-cytometry. The results showed a significant decrease in the percentage of CD133+ compartments in all the cells transfected with mim-486 or siRNA-p85 compared to cells transfected with ctrl-v (Figure 3C,D).

Flow-cytometry multiplex assays were also adopted to more deeply evaluate apoptosis in the CD133+ cell subpopulation. In the TP53 WT tumor cell lines H460, LT73, and A549 transfected with mim-486-5p and siRNA-p85, a higher percentage of CD133+ cells in early-stage apoptosis (7AAD-/Annexin+) was observed when considering both the bulk and the CD133+ population. Conversely, in the TP53 null H1299 cell line, induction of apoptosis was observed only when looking at the CD133+ subpopulation, suggesting that the effects on CD133+ cells are independent of the TP53 status (Figure 3E,F).

To verify a possible direct regulation of CD133 by miR-486-5p, experiments of luciferase reporter assay using commercial custom-made 3′ UTR CD133 were performed. The results did not show a down-modulation of the luciferase activity when HEK-293 cells were co-transfected with mim-486 compared to the control vector, allowing us to conclude that CD133 is not a direct target of miR-486-5p (Appendix A).

Altogether, our results indicate that miR-486-5p can affect CD133+ CSC by acting on p85 with consequent impairment of PI3K/Akt signaling activation.

### 2.4. miR-486 Overexpression Impairs In Vivo CD133+ Tumor-Initiating Properties

To better evaluate the in vivo impact of miR-486-5p on CD133+ CSCs, stable expression of miR-486 in A549 cells was generated using lentiviral vectors and overexpression of miR-486-5p was confirmed by RT-qPCR (Appendix A). Using this model, we initially estimated that the frequency of CSCs was significantly reduced (*p* < 0.0001) in cells overexpressing miR-486-5p compared to cells transfected with the ctrl-v by ELDA assay (Appendix A and Figure 4A). Moreover, considering only established xenografts, a marked reduced tumor growth was observed in tumors ensuing from miR-486-5p-transfected A549 cells (Figure 4B–D).

A549 CD133+ and CD133neg cells were then sorted and transiently transfected with mim-486 or the ctrl-v to evaluate the tumor take rate. Once injected subcutaneously in immunocompromised mice, 500 CD133+ cells transfected with the ctrl-v successfully grew as xenografts in two out of four (50%) cases, while mim-486-transfected cells completely abrogated (0%) in vivo CD133+ tumor-initiating properties (Figure 5A). Conversely, the same percentage of CD133− cells showed less tumor-initiating capability, and miR-486-5p had no effect on CD133− cell take rate (Figure 5B). On the other hand, considering only established xenograft, miR-486-5p overexpression affected CD133− in vivo growth, including when increasing the number of injected cells up to 100,000 (Figure 5C,D).

### 2.5. CCL-486 Treatment Decreases CD133+ Percentage and Inhibits Tumor Growth in NSCLC Patient-Derived Xenografts

The therapeutic potential of miR-486-5p on tumor growth was then assessed using NSCLC PDX models. PDXs samples from two different patients (PDX111 and PDX305) were initially dissociated to single cells and transiently transfected with mim-486 and ctrl-v. When re-injected subcutaneously in immunocompromised mice, we observed a delay in tumor growth in both of the models transfected with mim-486: 4 days and 18 days to reach half of the maximal volume in PDX111 and PDX305, respectively (Figure 6A).

To strengthen the role of miR-486-5p in modulating PDX growth, mim-486 and the ctrl-v were encapsulated in coated cationic lipid-nanoparticles specific for miRNA delivery. The CCL-486 or the CCL-ctrl compounds were administered intraperitoneally twice a week in mice carrying PDX111 or PDX305 tumor samples (Appendix A). At the end of the experiments, a 40% tumor volume reduction (*p* = 0.027) for PDX111 and a 50% tumor volume reduction (*p* = 0.049) for PDX305 was observed (Figure 6C,D). Of note, CCL-486 treatment resulted in increased miR-486-5p expression in about 30% of tumor cells, as appreciable by in situ hybridization (Figure 6E and Appendix A). On the other hand, a 50% and 40% reduction in CD133+ cells after 20 days and at the end of the experiment, respectively, was observed in CCL-486- compared to CCL-ctrl-treated tumors (Figure 6F). Moreover, higher miR-486-5p levels were appreciable also in lungs, liver, and spleen upon CCL-486 treatment compared to CCL-ctrl, while no major side effects were observed in these organs (Appendix A).

## 3. Discussion

In this project, we investigated the role of miR-486-5p in NSCLC models and in particular, for the first time, in the CSC compartment. Results indicated that replacement of miR-486-5p in lung cancer cell lines induced a decrease in CD133+ CSCs subset in both in vitro and in vivo models through the inhibition of the PI3K/Akt survivability pathway.

Starting from the characterization of paired tissue samples collected from lung cancer screening volunteers, although miR-486-5p was expressed at different levels in the normal lung tissue of heavy-smoker NSCLC patients, a general reduction was observed in the corresponding tumor tissue, irrespective of tumor stage [15,17]. Our results are in line with previously reported data showing that miR-486-5p was down-modulated in the tumor tissue and implicated in the control of the CSC phenotype of colorectal cancer patients and in liver cancer through Sirtuin 1 mRNA regulation [21,22]. In hepatocellular carcinoma, the replenishment of miR-486-5p blocks the IGF-1R pathway by acting on the downstream mediators such as mTOR, STAT3, and c-myc [23]. In the breast, miR-486-5p inhibits the epithelial-mesenchymal transition of cancer cells by reducing the expression of PIM1 and PTEN [24]. The repression of PIK3R1 by miR-486-5p is also described in renal cell carcinoma and leads to a reduction in tumor aggressiveness [25]. On the other side, in prostate cancer, miR-486-5p induces different oncogenic pathways through the negative regulation of tumor suppressors such as FOXO1, PTEN, and SMAD2 [26].

In vitro functional assays to evaluate the effects of miR-486-5p overexpression in lung cancer cell lines showed an induction of apoptosis and a reduced cell migration and invasion capability in TP53 WT models only, as already reported [15]. However, when moving to in vivo models, a drastic reduction in tumor growth was observed also in the H1299 TP53 null cell line transfected with mim-486. Of note, this difference was not due to an increased sensitivity of transfected cells to the lack of nutrients they underwent when injected subcutaneously in mice, as suggested by their capability to grow in serum-free medium. Thus, we searched for potential targets of miR-486-5p that could explain our in vivo results and focused our attention on the p85 regulatory subunit of the PI3K complex. In fact, this molecule is involved in stem cell regulation in glioma models, where the CD133–p85 interaction has a fundamental role and promotes tumorigenic ability of CSC by activating the pro-survival PI3K/Akt pathway [19].

In lung cancer, CD133+ tumor cells with stem-like properties, including higher ability of self-renewal, tumor initiation, and drug resistance, have been widely described [3,27,28]. Silencing of p85 in lung cancer cell lines in our models were thus performed. Indeed, using both a specific p85-directed siRNA or miR-486-5p, a marked decreased in CD133+ cells was observed. This reduction was mainly due to an induction of apoptosis in lung CD133+ cells and, conversely to what was observed in the tumor bulk, this was independent of TP53. This is of particular interest since CD133+ CSCs have been reported to be responsible for chemoresistance and particularly prone to dissemination [4]. Thus, miR-486-5p replacement may have an important clinical role in counteracting cancer relapse and metastasis formation, which are the principal causes of cancer-related death.

We cannot exclude the possibility that additional molecular mechanisms may be involved in these processes, and further studies are needed to finely dissect the miR-486-5p downstream effectors. Nonetheless, we proved the efficacy of a novel therapeutic approach by the delivery of lipid nanoparticles containing miR-486-5p. CCL-486 administered intraperitoneally reduced the growth of lung cancer PDXs in an immunocompromised mouse model. The slight decrement in tumor size could probably be attributed to the low delivery efficiency of our particles, as reported in our previous work [24]. To improve the anti-tumoral potential of our compounds, the addition of lung cancer-specific ligands to the surface could increase the delivery and consequently the therapeutic use of these compounds in lung cancer.

As far as the therapeutic aspect is concerned, the depletion of this very small subset (<1%) of highly tumorigenic CD133+ cells resulted in a marked reduction in both tumor uptake and tumor growth in vivo [3,29]. In recent years, several therapies directed against CSCs were investigated for cancer treatment; however only a few of them are proceeding through regulatory approval [30]. Targeting the main effectors or downstream effectors of key signaling pathways, such as the PI3K/Akt, is one of the alternative proposed approaches. Thus far, two PI3K inhibitors have reached phase 2 in clinical trials for the treatment of breast cancer [31]. The results of the single-arm phase 2trial (NCT01790932) enrolling triple-negative metastatic breast cancer treated with Buparlisib, a pan-class PI3K inhibitor drug, were recently published, and the authors conclude that no confirmed objective responses were observed [32]. This is also in line with what was observed in another clinical trial (NCT01277757), enrolling advanced breast cancer patients with PIK3CA/Akt1 or PTEN mutations or loss, where the AKT inhibitor, MK-2206, showed limited clinical activity [33]. Despite being obtained in a different model, these studies taken together indicated that PI3K/Akt-pathway inhibition alone may not be sufficient as a therapeutic strategy. Concerning miR-486-5p, it was also recently demonstrated that the suppression of the twinfilin actin-binding protein 1 (TWF1) induced by this miRNA improved the susceptibility to Cisplatin of lung cancer cell lines both in vitro and in vivo [34].

In conclusion, since miRNAs have revealed an interesting potential for cancer therapy in preclinical models, future approaches using the combination of neutral lipid nanoparticles encapsulating synthetic miRNAs with standard chemotherapy regimens could ameliorate the efficacy of the anticancer activity of these new drugs. Despite challenges such as the improvement of systemic delivery or the onset of resistance to chemotherapeutic drugs, the combination of miRNA and chemotherapy represents an innovative strategy for the management of lung cancer.

## 4. Materials and Methods

### 4.1. Tissue and Plasma Sample Collection

Tumor and paired normal lung tissue as well as plasma samples were collected from 25 and 20 NSCLC patients enrolled in the Istituto Nazionale dei Tumori–Istituto Europeo di Oncologia (INT–IEO) lung cancer screening trial and the Multicentre Italian Lung Detection (MILD) trial, respectively [35,36]. All subjects signed giving their informed consent and the study was performed according to the Declaration of Helsinki. All experimental protocols were approved by the Internal Review and Ethics Boards of the Fondazione IRCCS Istituto Nazionale Tumori of Milan (Milan, Italy).

### 4.2. RT-qPCR

From tissue samples, total RNA was extracted using Trizol^®^ (Thermo Fisher Scientific, Waltham, MA, USA) following the manufacturer’s instructions and quantified using the NanoDrop 2000 (Thermo Fisher Scientific, Waltham, MA, USA). For cultured cells, total RNA was isolated using *mir*Vana PARIS Kit (Thermo Fisher Scientific, Waltham, MA, USA) following the manufacturer’s instructions. MicroRNA expression was evaluated by Rt-qPCR starting from 20 ng of total RNA and using the TaqMan microRNA Reverse Transcription Kit (Thermo Fisher Scientific, Waltham, MA, USA) and a TaqMan RT Primer (Thermo Fisher Scientific, Waltham, MA, USA) specific for miR-486-5p (Assay ID 001278). The small nucleolar RNU48 (Assay ID 001006) was adopted as the housekeeping gene and the minor expressor as the calibrator. Gene expression was evaluated by the TaqMan Gene Expression Assay for PIK3R1 (Assay ID: Hs00933163_m1) using HPRT (Assay ID: Hs99999909_m1) as the housekeeping gene and the minor expressor as the calibrator. For the qPCR, the Applied Biosystems 7900 System (Thermo Fisher Scientific, Waltham, MA, USA) according to the manufacturer’s instructions was adopted.

### 4.3. In Silico miRNA Target Prediction

To identify miR-486-5p target mRNAs, we used the publicly available online miRWalk database [37], using the following filters: validated targets included in miRTarBase, predicted by TargetScan, and miRDB databases with a binding probability equal to 1 were considered.

### 4.4. Luciferase Assay

To evaluate the direct regulation of CD133 mRNA by miR-486-5p, HEK293 cells were transfected with CD133 3′UTR-containing vector (Switchgear Genomics, Carlsbad, CA, USA) in combination with the control vector (ctrl-v) and miR-486-5p mimic (mim-486). Cells were then cultured for 48 h and assayed with the Luciferase Reporter Assay System (Switchgear Genomics, Carlsbad, CA, USA).

### 4.5. Cell Lines and Culture Conditions

Three commercial human lung cancer cell lines (NCI-H460, A549, and H1299) were purchased from the American Type Culture Collection (ATCC). The LT73 cell line was established from a primary lung tumor of a 68-year old Caucasian male with lung adenocarcinoma operated in our Institution. Cancer cells were cultured in RPMI 1640 medium supplemented with 10% heat-inactivated fetal bovine serum (FBS) and 1% penicillin-streptomycin (Sigma-Aldrich, City of Saint Louis, MO, USA). For starvation experiments, RPMI 1640 FBS-free medium with 1% penicillin-streptomycin (Sigma-Aldrich, City of Saint Louis, MO, USA) was adopted.

### 4.6. Transient and Stable Transfection

Lung cancer cells were transiently transfected using *mir*Vana miRNA mimic, short interfering RNA (siRNA), and the control vector, using Lipofectamine^®^ 2000 (Thermo Fisher Scientific, Waltham, MA, USA), according to the manufacturer’s instructions. Stable transfection was performed using SMARTchoice lentiviral vector (Thermo Fisher Scientific, Waltham, MA, USA) according to the manufacturer’s instructions.

### 4.7. Proliferation, Apoptosis, and Cell Cycle Evaluation

For proliferation assay 2 × 10^5^ cells were seeded in a 12-well plate. The count of viable cells was performed after 72 and 120 h using trypan blue (Sigma-Aldrich, City of Saint Louis, MO, USA). Each experiment was performed in triplicate, considering as standard output the mean value ± standard deviation.

Apoptosis was quantified by the percentage of Annexin V^pos^/7AAD^neg^ cells using flow-cytometry. Following the manufacturer’s instructions, the Annexin V Kit (Miltenyi Biotec, Bergisch Gladbach, Germany) was adopted to evaluate the percentage of apoptotic cells 48 h after miRNA transfection.

Cell cycle was evaluated 48 h after miRNA transfection. After fixation with 70% cold ethanol, cells were stained with propidium iodide (50 µg/mL) for 40 min. Analyses were carried out by flow-cytometry using BD FACS Calibur and Cell Quest software (BD Biosciences, Franklin Lakes, NJ, USA).

### 4.8. Migration and Invasion Assay

For migration assay, 10^5^ cells were placed on the top chamber of FluoroBlok Cell Culture Inserts (BD Biosciences, Franklin Lakes, NJ, USA). RPMI plus 10% FBS was used as stimulus in the bottom chamber and incubated at 37 °C and 5% CO_2_. For the invasion assay, filters were coated with matrigel (BD Biosciences, Franklin Lakes, NJ, USA). After 24 h, migrated cells on the bottom side of the filters were stained with DAPI and counted using fluorescence microscopy. Migration and invasion data are expressed as the number of migrated miR-486-5p overexpressing cells vs. the number of migrated cells transfected with the control vector. Each experiment was performed in triplicate.

### 4.9. In Situ Hybridization

miRNA In Situ Hybridization (ISH) was performed on 5 µm-thick FFPE tissue sections derived from patients, PDXs, and xenografts. Sections were first deparaffinized in xylene, rehydrated on alcohol descending scale, and treated with Proteinase-K (Sigma-Aldrich, City of Saint Louis, MO, USA), diluted 1:200, for 30 min at 37 °C on a Dako Hybridizer (Dako, Fort Collins, CO, USA). Slides were then washed in PBS 1X, dehydrated in alcohol increasing scale and hybridized with miRNA detection probes. We used double digoxigenin (DIG)-conjugated mirCURY LNA detection probes from Exiqon—a specific hsa-miR-486-5p probe (5’TCGGGGCAGCTCAGTACAGGA; RNA-T_m_: 84.5°C) and a scrambled (negative) probe (GTGTAACACGTCTATACGCCCA; RNA-T_m_: 87.3 °C)—at final concentrations of 100 nM and 40 nM, respectively. For PDX and xenograft tissue sections, miRNA detection was performed using Tyramide Signal Amplification (TSA) Plus Kit (Perkin Elmer, Waltham, MA, USA) and Anti-digoxigenin (mouse)—HRP conjugate (Perkin Elmer, Waltham, MA, USA), diluted 1:1000, according to the manufacturer’s instruction.

For patients, tissue section miRNA expression was automatically detected with the Ventana BenchMark ULTRA instrument (Ventana Medical Systems, Oro Valley, AZ, USA) using the OptiviewDAB Detection Kit (Ventana Medical Systems, Oro Valley, AZ, USA), as previously described [38]. Stained sections were examined by optical microscope and scanned with Aperio Scanscope XT (Leica Biosystems, Wetzlar, Germany).

### 4.10. Western-Blot Analysis

Protein extraction was performed using RIPA buffer (Sigma-Aldrich, City of Saint Louis, MO, USA) and total lysate was quantified by Bradford reagent. Twenty-five micrograms of protein was separated on Nupage 4–12% polyacrylamide gels (Thermo Fisher Scientific, Waltham, MA, USA) and transferred to polyvinylidene difluoride membranes (GE Healthcare, Chicago, IL, USA). The membranes were blocked with 5% milk in phosphate-buffered saline with 0.05% Tween 20 (PBS-T) buffer to be probed with the following antibodies: Anti-PI3K p85 (mAb4292, Cell Signaling, Danvers, MA, USA), Phospho-Akt (Ser473) (mAb4060, Cell Signaling, Danvers, MA, USA), Akt (pan) (mAb4691, Cell Signaling, Danvers, MA, USA), and rabbit anti-β-actin (Sigma-Aldrich, City of Saint Louis, MO, USA), by standard protocol. For detection, goat anti-rabbit or goat anti-mouse secondary antibodies conjugated to horseradish peroxidase (GE Healthcare, Chicago, IL, USA) were used accordingly. Signal detection was performed via chemiluminescence reaction (GE Healthcare, Chicago, IL, USA).

### 4.11. Immuno-Fluorescence

Immuno-fluorescence experiments on the LT73 tumor tissue and the derived lung cancer cell line were performed using rabbit Anti-PI3K p85 (mAb4292, Cell Signaling, Danvers, MA, USA) and mouse CD133/1 Antibody (130–113-670, Miltenyi Biotec, Bergisch Gladbach, Germany), according to the manufacturer’s instructions. Cells were blocked using PBS with 3% BSA and 2% normal goat serum for 30 min at room temperature. After being washed twice with 1% BSA, cells were co-incubated overnight at 4°C with primary antibodies diluted 1:20 in 1% BSA. After being washed 3 times with PBS, cells were then co-incubated for up to 4 h with goat anti-rabbit Alexa 488 IgG (Invitrogen, Waltham, MA, USA) and goat anti-mouse Alexa 594 IgG (Invitrogen, Waltham, MA, USA) diluted 1:1000 in 1% BSA. Nuclei were stained with the 4’,6-diamidino-2-phenylindole (DAPI) fluorescence. Images were uptaken with the fluorescent microscope Olympus BX51 and Software MacProbe v4.2.3.

### 4.12. Fluorescence-Activated Cell Sorting (FACS + SORTING)

Lung cancer cells were incubated with anti-human PE-CD133/1 (Clone AC133, Miltenyi Biotec, Bergisch Gladbach, Germany), and diluted 1:50 in PBS+ 0.5% BSA for 30 min at 4 °C cells. FACSCalibur and FACSCanto cell analyzers (BD Biosciences, Franklin Lakes, NJ, USA) were used for data acquisition and FlowJo software V10 for data analysis. To obtain purified CD133+ and CD133− cells for in vivo tumorigenic assay, A549 cell line was sorted for CD133 using FACSAria cell sorter (BD Biosciences, Franklin Lakes, NJ, USA), as already reported [39].

### 4.13. Extreme Limiting Dilution Assay (Con Referenza Da Sito)

Extreme Limiting Dilution Assay (ELDA) was performed adapting the protocol described by Drs. Hu and Smyth to establish the effective number of CSCs [40]. In particular, scalar doses of mim-486- or ctrl-v-stably transfected A549 cells (10,000, 100,000, or 500,000) were injected subcutaneously in the right flank of 12 mice and tumor take was annotated. The 1/(stem cell frequency) value was estimated based on the tumor take rate in each condition.

### 4.14. In Vivo Experiments (Xenografts and PDX)

All xenograft experiments were carried out using 8- to 9-week-old female SCID mice (from Charles River Laboratories, Calco, Italy) maintained in laminar-flow rooms at constant temperature and humidity, with food and water given ad libitum. Before implantation in mice, cells were tested for the presence of rodent pathogens using IMPACT I PCR Profile (IDEXX BioResearch, Columbia, MO, USA) and resulted negative. Tumor cells (5 × 10^5^ cells/mouse) were injected subcutaneously (s.c.) on the mouse’s right flank. Tumor mass was measured with a caliper and tumor volume (mm^3^) calculated (long diameter × (short diameter)^2^/2).

PDXs were established as previously described [20]. Experiments were carried out in groups of 8 mice, bearing a PDX sample in the right flank. Mice were treated twice a week with intraperitoneal injection, with 1.5 mg/Kg of the miR-486-5p- or miR-SCR-entrapped lipid-nanoparticles (CCL-486 or CCL-SCR, respectively) or HEPES-buffered saline as the non-treated (N.T), twice a week for four weeks. Tumor growth was followed by caliper twice a week and results were analyzed using GraphPad Prism software. Distribution of the CCL-486 compound into mice organs was evaluated by QuantStudio™ 3D Digital PCR (dPCR) using Taqman MicroRNA assays (Thermo Fisher Scientific, Waltham, MA, USA), miR-486-5p (ID:001278), and cel-mir-39 (CUSTOM) as previously described [41]. All the experimental protocols were carried out according to the Italian law D.Lgs. 26/2014, and animal experimentation was performed following guidelines drawn up by Fondazione IRCCS Istituto Nazionale dei Tumori Institutional Animal Welfare Body [42].

## Figures and Tables

**Figure 1 pharmaceuticals-15-00297-f001:**
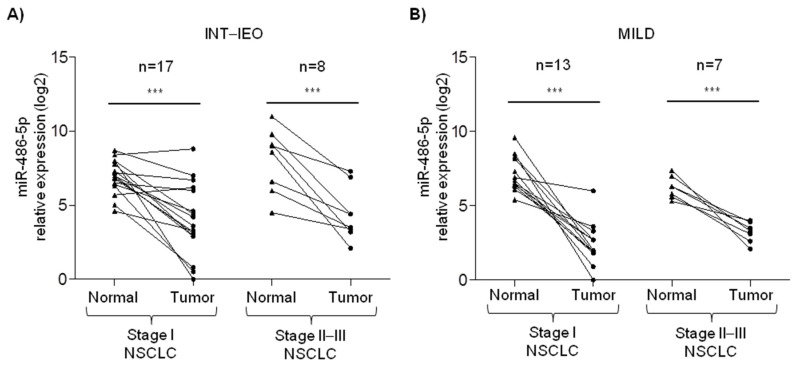
miR-486-5p expression in paired normal and lung tumor tissue. RT-qPCR analysis of miR-486-5p relative expression normal lung and paired tumor tissue samples collected from stage I and stage II–III lung cancer patients identified during (**A**) the Istituto Nazionale dei Tumori–Istituto Europeo di Oncologia (INT–IEO) and (**B**) the Multicentre Italian Lung Detection (MILD) lung cancer screening trials. Asterisks represent the level of significance by paired *t*-test *p*-value (*** <0.001).

**Figure 2 pharmaceuticals-15-00297-f002:**
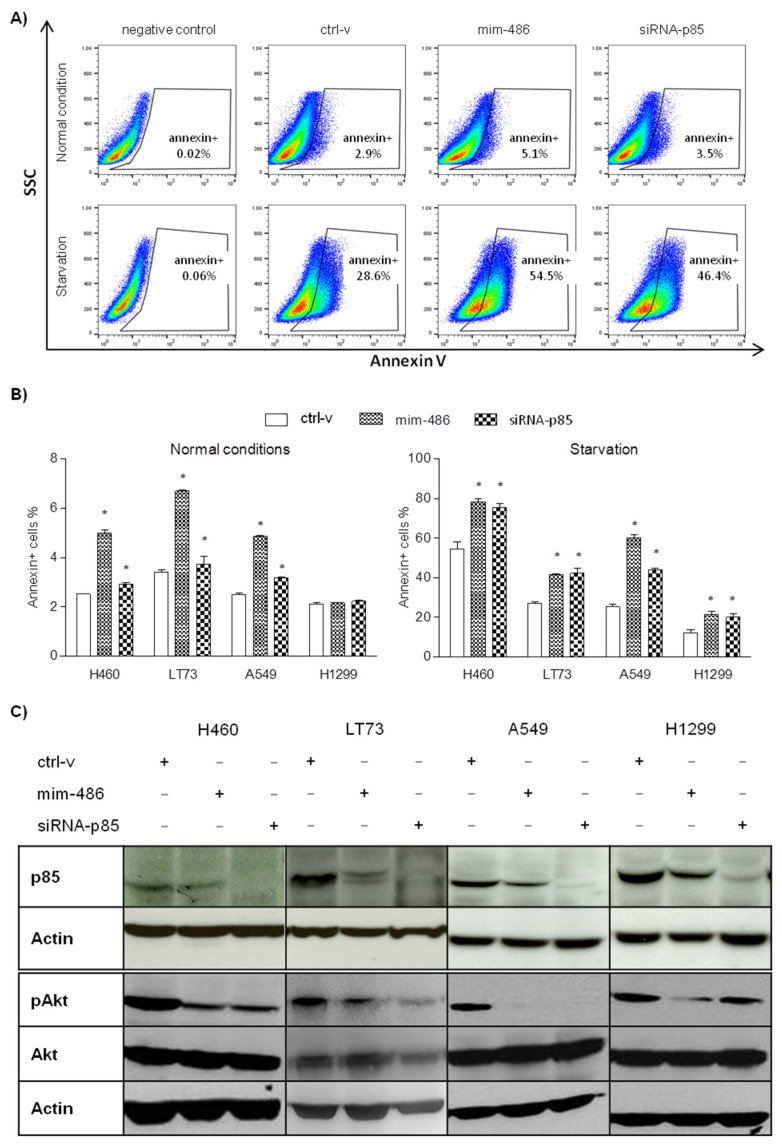
miR-486-5p induced cell survivability of lung cancer cell lines regulating the p85/AKT pathway. (**A**) In A549 and (**B**) other lung cancer cell lines cultured in both normal and starved conditions, the mim-486 and siRNA-p85 transfection induced early apoptosis, evaluated as percentage of Annexin+ cells by flow-cytometry. (**C**) Western blot of H460, LT73, A549, and H1299 cell lines transfected with mim-486, siRNA-p85, and the control vector (ctrl-v) to evaluate the p85/AKT pathway activation. Actin was used as reference to normalize data. *: *p* < 0.05 vs ctrl-v.

**Figure 3 pharmaceuticals-15-00297-f003:**
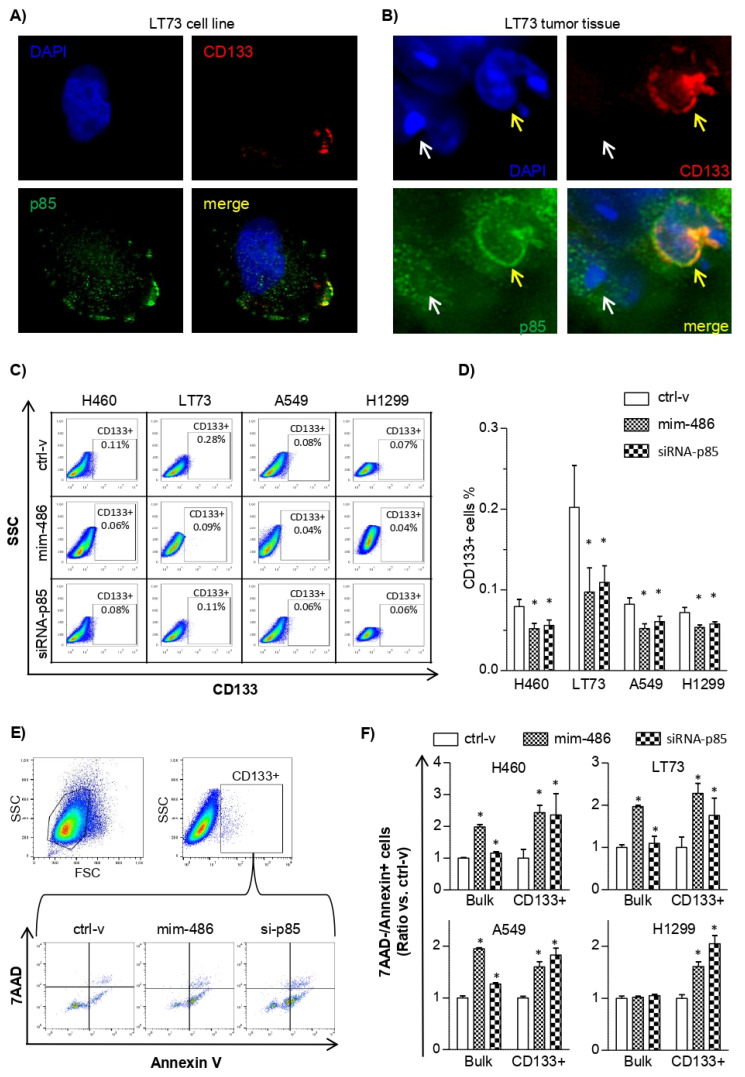
miR-486-5p reduces CD133+ lung cancer stem cells through the regulation of the p85/AKT pathway. (**A**,**B**) Immuno-fluorescence assay evidences the co-localization of CD133 (PI, red) and p85 (FITC, green) on the membrane of CD133+ lung cancer stem cells in the LT73 cell line and in the LT73 original tumor tissue. In panel B, CD133+ cell is indicated by the yellow arrow and CD133– cell by the white arrow. (**C**,**D**) Flow-cytometry analysis of CD133+ cells in lung cancer cell lines transfected with the control vector (ctrl-v), mim-486, and siRNA-p85. (**E**) Gating strategy to evaluate 7AAD and Annexin expression in LT73 CD133+ cell lines. (**F**) Percentage of Annexin+ cells in lung cancer cell lines transiently transfected with the control vector (ctrl-v), mim-486, and siRNA-p85. For all the experiments, 48 h after transient transfection was selected as time point. *: *p* < 0.05 vs ctrl-v.

**Figure 4 pharmaceuticals-15-00297-f004:**
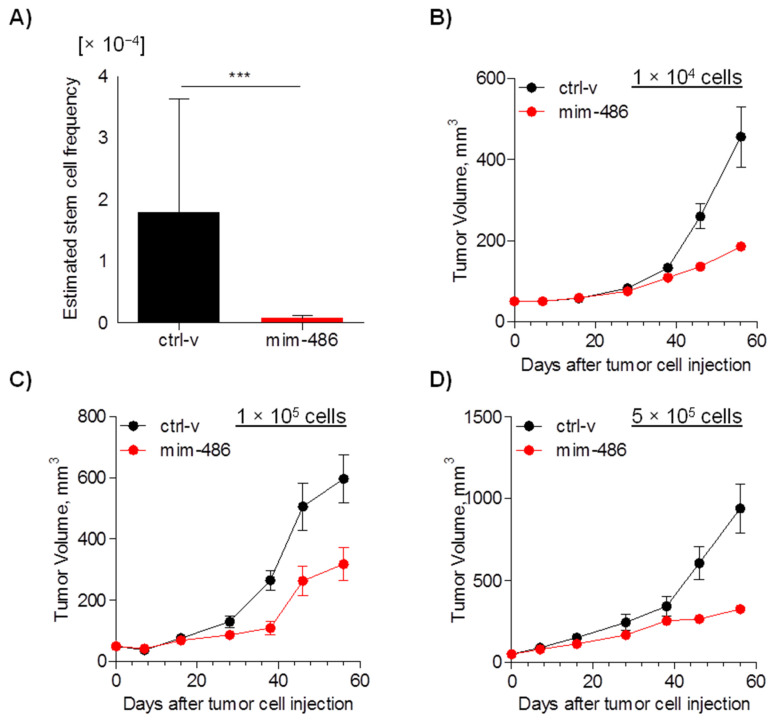
miR-486-5p reduces cancer stem cell frequency in vivo. (**A**) Estimated cancer stem cell frequency evaluated by Extreme Limiting Dilution Assay (ELDA) in A549 cells transfected with mim-486 or control vector. (**B**–**D**) Growth curves of A549 cells injected subcutaneously in SCID mice at 1 × 10^4^, 1 × 10^5^, and 5 × 10^5^ cells. ELDA data are expressed as median and interval between minimum and maximum; in vivo growth data are expressed as mean ± standard error of the mean (SEM).

**Figure 5 pharmaceuticals-15-00297-f005:**
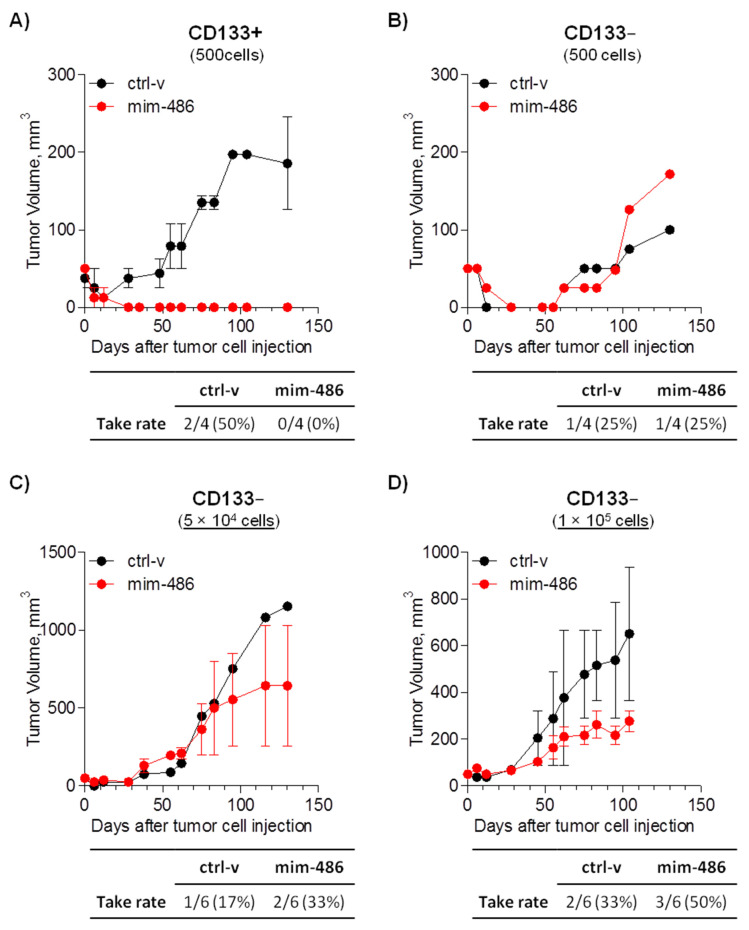
miR-486-5p reduces in vivo tumor take of A549 CD133+ cells. (**A**) Take rate and growth curve of 500 CD133+ cells transfected with the control vector (ctrl-v) or miR-486-5p mimic (mim-486) and injected subcutaneously in SCID mice. Take rate and growth curve of (**B**) 500, (**C**) 5 × 10^4^, and (**D**) 1 × 10^5^ CD133− cells transfected with the control vector (ctrl-v) or miR-486-5p mimic (mim-486) and injected subcutaneously in SCID mice. Growth curves were obtained considering only established xenograft. All data are expressed as mean ± standard error of the mean.

**Figure 6 pharmaceuticals-15-00297-f006:**
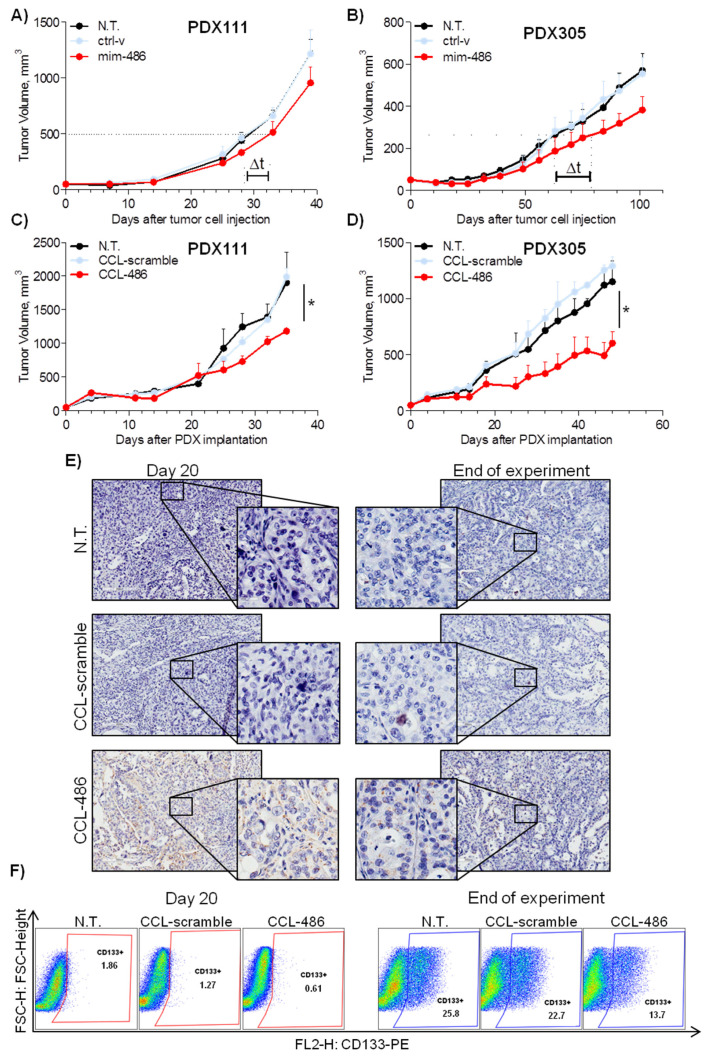
miR-486 reduces in vivo tumor growth and CD133+ cell percentage in PDXs. (**A**,**B**) Cells derived from two different PDXs (PDX111 and PDX305) and transfected with mim-486 show a delay in growth (Δt) compared to non-transfected or control vector-transfected cells when injected subcutaneously in SCID mice. (**C**,**D**) CCL-486 administered intraperitoneally twice a week significantly decreases growth of PDX111 and PDX305 injected subcutaneously in SCID mice compared to untreated and control vector-treated PDXs. (**E**) In situ hybridization (ISH) analysis indicates miR-486 expression only in CCL-486-treated PDXs both after 20 days and at the end of the experiment. (**F**) Representative FACS analysis indicates a reduction in CD133+ cells levels in CCL-486-treated PDXs compared to those in untreated and control vector-treated PDXs, both after 20 days and at the end of the experiment. All in vivo growth data are expressed as mean ± standard error of the mean (SEM).

**Table 1 pharmaceuticals-15-00297-t001:** Clinico-pathological characteristics of lung cancer screening patients.

Characteristics	INT–IEO (tot = 25)		MILD (tot = 20)	
	N	%	N	%
Gender				
Females	7	28	5	25
Age				
Mean (range)	59 (50–72)		62 (52–74)	
Pack-year				
Mean (range)	61 (26–100)		61 (27–127)	
Tumor histology				
Adenocarcinoma	19	76	14	70
Others	6	24	6	30
Tumor stage				
Ia–Ib	17	68	13	65
II–III	8	32	7	35

## Data Availability

Data is contained within the article and Appendix A.

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
