# Peer review of "MiR-486-5p Targets CD133+ Lung Cancer Stem Cells through the p85/AKT Pathway"

_pharmaceuticals, 2022, doi:10.3390/ph15030297_

Round 1
Reviewer 1 Report
The article is devoted to the role and potential therapeutic use of miRNA-486-5p in lung cancer, emphasising the molecular events and impact on CD133-positive cancer stem cells. The main strengths of the study are represented by its topicality, appropriate experimental level and comprehensibility. The scientific team has targeted several interlinked hot topics: miRNA, cancer stem cells and a tumour - namely, lung cancer - that is well-known for insufficiently effective treatment. In my opinion, the study is timely, original and significant; it is written on high scientific level.
Among the few critical remarks, the manuscript should be checked for the rare misprints ("trough" instead of "through", "transfetcted" in the legend of Figure 4, etc.) and formatting errors (e.g., few missing e-mail addresses).
From the scientific point of view, the role of miRNA-486-5p in physiological processes and other tumours could be shortly described in the Discussion. This would help to understand if therapeutic manipulation of miRNA levels might have any predictable side effects.
The other molecular pathways triggered by miRNA-486-5p could also be shortly mentioned in the Discussion.
Author Response
Manuscript ID: pharmaceuticals-1559875, entitled "Mir-486-5p targets CD133+ lung cancer stem cells through the p85/AKT pathway"
Response to Reviewers
We are submitting to your attention the revised version of the article “Mir-486-5p targets CD133+ lung cancer stem cells through the p85/AKT pathway ".
We would like to thank both reviewers for the accurate revisions and important insights provided on the first version of the manuscript. We have tried to follow all suggestions, which have greatly helped us in improving the quality of the paper.
In detail, we present here the point-by-point response to reviewer 1 comments/suggestions:
REVIEWER 1
The article is devoted to the role and potential therapeutic use of miRNA-486-5p in lung cancer, emphasising the molecular events and impact on CD133-positive cancer stem cells. The main strengths of the study are represented by its topicality, appropriate experimental level and comprehensibility. The scientific team has targeted several interlinked hot topics: miRNA, cancer stem cells and a tumour - namely, lung cancer - that is well-known for insufficiently effective treatment. In my opinion, the study is timely, original and significant; it is written on high scientific level.
Among the few critical remarks, the manuscript should be checked for the rare misprints ("trough" instead of "through", "transfetcted" in the legend of Figure 4, etc.) and formatting errors (e.g., few missing e-mail addresses).
A1. We are sorry for the errors and typos we made. Misprints have been checked throughout the manuscript.
From the scientific point of view, the role of miRNA-486-5p in physiological processes and other tumours could be shortly described in the Discussion. This would help to understand if therapeutic manipulation of miRNA levels might have any predictable side effects.
The other molecular pathways triggered by miRNA-486-5p could also be shortly mentioned in the Discussion.
A2. The following paragraph and sentence were included in the Discussion session to meet the Reviewer's request:
“Starting from the characterization of tissue samples collected from lung cancer screening volunteers, we observed a reduction of miR-486-5p expression in tumor tissue of NSCLC patients [15,17]. Our results are in line with previously reported data showing that miR-486-5p was down-modulated in the tumor tissue and implicated in the control of the CSC phenotype of colorectal cancer patients and in liver cancer through Sirtuin 1 mRNA regulation [21,22]. In hepatocellular carcinoma, the replenishment of miR-486-5p blocks the IGF-1R pathway by acting on the downstream mediators such as mTOR, STAT3 and c-myc [23]. In breast, miR-486-5p inhibits the epithelial-mesenchymal transi-tion of cancer cells by reducing the expression of PIM1 and PTEN [24]. The repression of PIK3R1 by miR-486-5p is also described in renal cell carcinoma and lead to a reduction of tumor aggressiveness [25]. On the other side, in prostate cancer miR-486-5p induces different oncogenic pathways through the negative regulation of tumor suppressors such as FOXO1, PTEN and SMAD2 [26]”
“Concerning miR-486-5p, it was also recently demonstrated that the suppression of the twinfilin actin-binding protein 1 (TWF1) induced by this miRNA improved the susceptibility to Cisplatin of a lung cancer cell lines both in vitro and in vivo [34].”
Reviewer 2 Report
Major concern
- Why the relative expression levels of miR-486-5p in normal lung tissue between patients at stage 1 and those at stage more than 1 are different?
- What kind of information that figure 1b try to deliver? Please describe the rationale for collecting plasma from lung cancer patients and analyzing the miR-486-5p expression correlation with normal/tumor lung data.
- Although the evidence showed the serum starvation could activate the Akt pathway of cells, it will induce an extra survival signal pathway of cell lines. Therefore, the authors can not conclude that action of miR-486-5p is directly through the Akt pathway.
- It is not proper to conclude that CD133 colocalized with p85 in just a few cells images (figure 3A and B).
- In Figures 3c and d, the authors concluded that siRNA-p85 significantly decreased the percentage of CD133+ cells. However, the data provided shows only 0.28% to 0.09%/0.11% difference for the LT73 cell line. It is highly recommended that the authors analyze the expression profile of miR-486-5p, p85, and CD133 in the native state of each tumor cell line.
- Is it proper to use ELDA to analyze CD133 cell occurring frequency? How about using the flow to demonstrate it?
Author Response
Manuscript ID: pharmaceuticals-1559875, entitled "Mir-486-5p targets CD133+ lung cancer stem cells through the p85/AKT pathway"
Response to Reviewers
We are submitting to your attention the revised version of the article “Mir-486-5p targets CD133+ lung cancer stem cells through the p85/AKT pathway ".
We would like to thank both reviewers for the accurate revisions and important insights provided on the first version of the manuscript. We have tried to follow all suggestions, which have greatly helped us in improving the quality of the paper.
In detail, we present here the point-by-point response to reviewer 2 comments/suggestions:
REVIEWER 2
Major concern
- Why the relative expression levels of miR-486-5p in normal lung tissue between patients at stage 1 and those at stage more than 1 are different?
A1. As highlighted by the Reviewer, the difference between normal lung tissue in stage I vs. stage>I was statistically significant (p=0.03). This difference was mainly due to a couple of outliers exceeding the mean value + standard deviation of the miR-486-5p expression levels in the whole series. Of note, one of these 2 outliers is the only small cell lung cancer (SCLC) patient identified by the screening program. By excluding this sample, the difference is no more significant (p=0.09). On the other hand, since there is only one SCLC patients, we cannot conclude that the difference is due to tumor histology.
To avoid any confusion we prefer to exclude this sample and focus only on NSCLC patients. Moreover, we have now included data obtained by the analysis of an additional series of normal and tumor tissue samples collected from 20 NSCLC patients enrolled in a second independent LDCT screening trial. Even in this second series no differences were found when comparing miR-486-5p expression levels in normal lung tissue of stage I vs. stage>I tumors (p=0.09). The main text, Figure 1 and Table 1 were accordingly modified.
2. What kind of information that figure 1b try to deliver? Please describe the rationale for collecting plasma from lung cancer patients and analyzing the miR-486-5p expression correlation with normal/tumor lung data.
A2. Levels of microRNA in plasma are routinely analysed in our lab with the aim to develop and validate a minimally invasive diagnostic tool. However, given the therapeutic implication of the present work, we agree with the Reviewer that this analysis is indeed off topic and therefore Figure 1b and the relative text have been removed from the new version of our paper.
3. Although the evidence showed the serum starvation could activate the Akt pathway of cells, it will induce an extra survival signal pathway of cell lines. Therefore, the authors can not conclude that action of miR-486-5p is directly through the Akt pathway.
A3. As now better reported in the Introduction and Discussion of our paper, miR-486-5p was demonstrated to regulate several targets (see Answer 2 to Reviewer 1). Among the others, a direct role of miR-486-5p in the regulation the Akt pathway was also described. Moreover, here we confirmed that the transfection of miR-486-5p mimic in cancer cell lines affect the phosphorilation of Akt as much as the silencing of p85 (fig.2C). On the other hand, we cannot conclude that the action of miR-486-5p is directly exclusively through the Akt pathway as now better discussed: “We cannot exclude that additional molecular mechanisms can be involved in these processes and further studies are needed to finely dissect the miR-486-5p downstream players”.
4. It is not proper to conclude that CD133 colocalized with p85 in just a few cells images (figure 3A and B).
A4. The paragraph describing the co-localization of CD133 and p85 was modified to tone down data interpretation, as suggested by the Reviewer: “(…) immuno-fluorescence experiments, carried out on the primary adenocarcinoma tissue and in the corresponding LT73 established cell line, indicate a possible co-localization of p85 and CD133 at the level of the cellular membrane in lung CD133+ cells (Figure 3A,B). Conversely, in CD133neg cells p85 localizes exclusively in the cytoplasm”.
5. In Figures 3c and d, the authors concluded that siRNA-p85 significantly decreased the percentage of CD133+ cells. However, the data provided shows only 0.28% to 0.09%/0.11% difference for the LT73 cell line. It is highly recommended that the authors analyze the expression profile of miR-486-5p, p85, and CD133 in the native state of each tumor cell line.
A5. To verify the role of miR-486 in the control of CSC phenotype we profile miRNA, p85 and CD133 cells in all the 4 cell lines. The results reported in Supplementary Figure S3 suggested that miR-486 is associated to p85 and CD133 expression. The following sentence was also included in the Results section: “The expression profiles of miR-486-5p and p85 in the 4 parental lung cancer cell lines, as well as the amount of CD133+ cells evaluated by flow-cytometry, suggest the presence of the cross-talk between miR-486-5p/p85 axis and the CSC phenotype also in these models (Supplementary Figure S3)”.
6. Is it proper to use ELDA to analyze CD133 cell occurring frequency? How about using the flow to demonstrate it?
A6. ELDA was used to functionally estimate the frequency of cancer stem cells (CSCs) based on their potential to initiate a tumor in vivo. Here, we exploited ELDA to test whether the modulation of CSC caused by mir-486 transient transfection, as evaluated by FACS (Fig3C-D), was indeed associated to a decrease in tumor take rate between ctrl-v and mim-486 transfected cells. The results demonstrated that effects of mir-486 on CSC are reflected by a different take rate.
The amount of CSC, involved in the initial phase of tumour initiation, is expected to revert to a “physiological” percentage of CD133+ cells similarly in all grown xenografts, since this cell subset is able to reform the tumor heterogeneity (as demonstrated in Bertolini et al. PNAS 2009). Thus, a difference in CSCs frequency evaluated by FACS in ctr-v and mim-486 grown tumors was not expected in this kind of experiment.
Round 2
Reviewer 2 Report
- Why the relative expression levels of miR-486-5p in normal lung tissue between patients at stage 1 and those at stage more than 1 are different?
As the authors’ response, “Of note, one of these 2 outliers is the only small cell lung cancer (SCLC) patient identified by the screening program.”. However, theoretically, the relative expression levels of miR-486-5p for the normal lung tissue shall be the same no matter the patient’s cancer situation. Especially in version 2 the author included another group of patient samples (MILD) which the relative expression levels of miR-486-5p in normal tissue are lower than INT-IEO. The authors shall discuss this in the discussion part.
- In the supplementary data (Figure S2), it seems the mim-486 expression levels are not highly correlated with cell growth, apoptosis, migration, invasion, and in vivo tumor growth. It seems not to obey the rationale of this study. Please explain the possible reason for this observation.
- In section 2.4, the authors try to demonstrate the miR-486 overexpression impairs CD133+ CSCs’ function. Although the results of ELDA shown in figure 4 concluded overexpression of miR-486 markedly reduced tumor growth, the correlation of miR-486 expression and CD133+ population shown in figure S1and figure 3 are not evident in this hypothesis, especially in the LT73 cell line.
- The hypothesis of this study emphasizes that miR-486 might target CD133 (figure S4). Since the microRNA affects the protein expression, please explain why the CD133+ CSC cells would be affected by miR-486 transfection instead of non-CSC cell in Figure5A and B.
Author Response
Thank you for your review, Please see the attachment for the authors' response.

Round 3
Reviewer 2 Report
All my questions have been properly answered.